# Adaptive Multi-prompt Contrastive Network
# for Few-shot Out-of-distribution Detection

**Xiang Fang** [1 2]  **Arvind Easwaran** [2]  **Blaise Genest** [3]

## Abstract

Out-of-distribution (OOD) detection attempts to distinguish outlier samples to prevent models trained on the in-distribution (ID) dataset from producing unavailable outputs. Most OOD detection methods require many ID samples for training, which seriously limits their real-world applications. To this end, we target a challenging setting: few-shot OOD detection, where only a few *labeled ID* samples are available. Therefore, few-shot OOD detection is much more challenging than the traditional OOD detection setting. Previous few-shot OOD detection works ignore the distinct diversity between different classes. In this paper, we propose a novel network: Adaptive Multi-prompt Contrastive Network (AMCN), which adapts the ID-OOD separation boundary by learning inter- and intra-class distribution. To compensate for the absence of OOD and scarcity of ID *image samples*, we leverage CLIP, connecting text with images, engineering learnable ID and OOD *textual prompts*. Specifically, we first generate adaptive prompts (learnable ID prompts, label-fixed OOD prompts and label-adaptive OOD prompts). Then, we generate an adaptive class boundary for each class by introducing a class-wise threshold. Finally, we propose a prompt-guided ID-OOD separation module to control the margin between ID and OOD prompts. Experimental results show that AMCN outperforms other state-of-the-art works.

## 1. Introduction

Deep neural networks (DNNs) receive more and more attention due to their wide machine learning applications, such as image classification (Bai et al., 2024; Gu et al., 2024; Li et al., 2024a). Unfortunately, most of DNNs refer to a closed-set assumption that all the test samples are seen during training and no outliers are observed during inference. In fact, there are many unseen test samples (*i.e.*, outliers) in real-world applications, such as autonomous driving (Zendel et al., 2022; Vyas et al., 2018; Lu et al., 2023). These DNN methods still mistakenly classify each outlier into a seen class. The wrong classification of outliers will result in irrecoverable losses in some safety-critical scenarios. To solve the above problem, the out-of-distribution detection (OOD detection) task (Gautam et al., 2023; Zhang et al., 2024a; Hendrycks & Gimpel, 2016; Sun & Li, 2022; Fort et al., 2021; Liu et al., 2020) is proposed to accurately detect outliers in OOD classes and correctly classify samples from in-distribution (ID) classes during testing. Therefore, OOD detection has attracted increasing attention and various OOD detection models have been proposed for various safety-critical scenarios (Kirchheim et al., 2024; Abrecht et al., 2024; Kaur et al., 2024; Wu et al., 2024). Most OOD detection works (Shen et al., 2024; Regmi et al., 2024a; Xue et al., 2024; Regmi et al., 2024b; Zhang et al., 2024b) refer to a fully-supervised assumption that samples of all types (*e.g.,* all races of cats) of an ID class (*e.g.,* cat) in all situations are accessible during training, which is unrealistic. Further, many OOD samples are also usually required during training.

*Few-shot OOD detection* is posed to first train the designed model on a few samples in each ID class and then conduct OOD detection on the whole test set. Such a setting limits the performance of standard OOD detection methods. Existing few-shot OOD detection task meets the following challenges: 1) Most few-shot methods (Jeong & Kim, 2020; Dionelis et al., 2022; Mehta et al., 2024; Zhan et al., 2022; Zhu et al., 2024) are sensitive to the background of the image. When they train the designed model on a few number of images of each class, it might lead to the understanding bias, resulting in the wrong OOD detection results. For example, "dog" (ID) class and "wolf" (OOD) class share many visual characteristics. When a dog appears on the grassland, the designed model might misidentify it as a wolf. Besides, some similar classes have various backgrounds, which might lead to wrong OOD detection reasoning results. For instance, in a real-world training dataset, cat images are mainly indoors, while dog images are mostly outdoors. Thus, the

---

[1]Energy Research Institute @ NTU, Interdisciplinary Graduate Programme, Nanyang Technological University, Singapore [2]College of Computing and Data Science, Nanyang Technological University, Singapore [3]CNRS and CNRS@CREATE, IPAL IRL 2955, France and Singapore. Correspondence to: Xiang Fang <xiang003@e.ntu.edu.sg>.

*Proceedings of the $42^{nd}$ International Conference on Machine Learning*, Vancouver, Canada. PMLR 267, 2025. Copyright 2025 by the author(s).

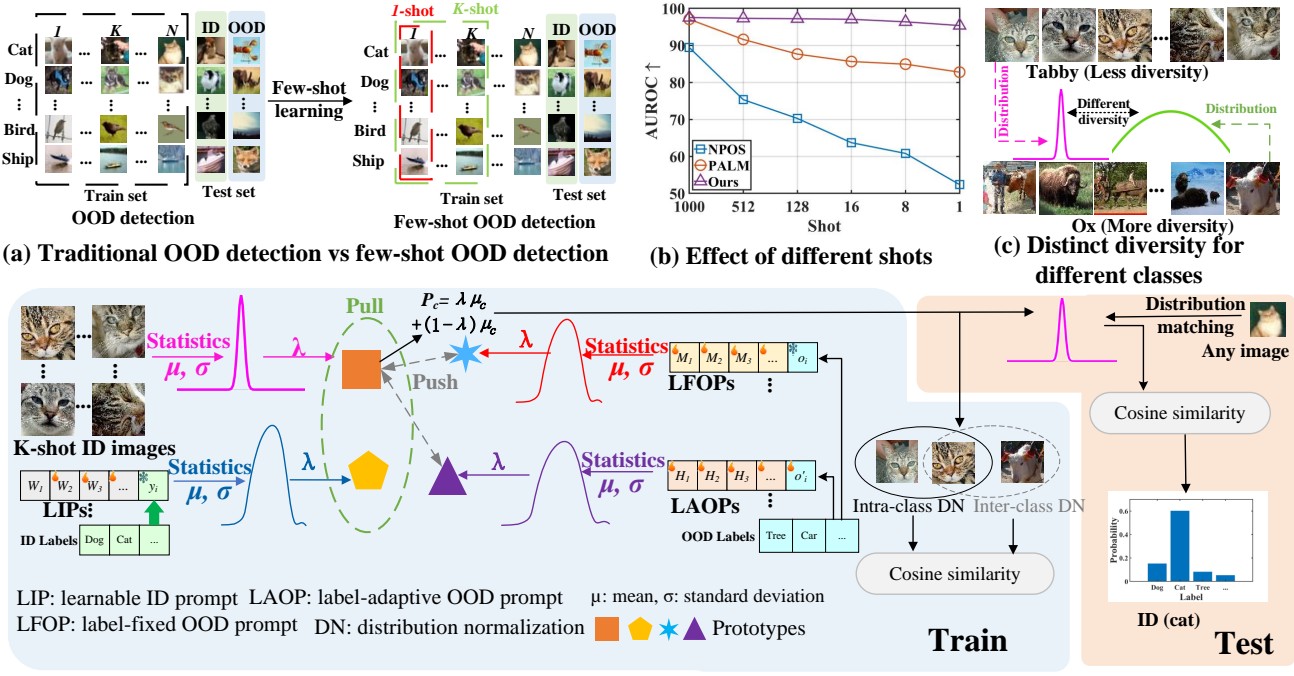

**(a) Traditional OOD detection vs few-shot OOD detection**

**(b) Effect of different shots**

**(c) Distinct diversity for different classes**

**(d) Framework of our proposed method**

*Figure 1.* (a) Comparison between traditional OOD detection and Few-shot OOD detection. (b) Performance of different methods (NPOS (Tao et al., 2023), PALM (Lu et al., 2024) and Ours) on the SUN dataset with ImageNet-1k as the train set. (c) Diversity comparison between the "cat" class and the "ox" class. (d) Brief framework of our method. Best viewed in color.

dog images often contain more complex backgrounds than the cat images. Besides, different classes have various levels of diversity, which makes it difficult to accurately learn the class boundaries with only a few samples. 2) In the few-shot setting, as the class number increases, the model performance always decreases since the ID-OOD boundary becomes more complex. Thus, the model has to learn more subtle differences between classes with only a few examples. With more classes, there is a greater likelihood of overlap between class features, making it harder for the model to distinguish between them accurately. 3) Few-shot learning inherently suffers from accessing limited samples, which exacerbates the issue of lacking representative examples to generalize well across more classes. Also, the limited samples might lead to overfitting, seriously limiting the model performance.

To address the above challenges, we design a novel network for the challenging few-shot OOD detection task. 1) To compensate for the absence of OOD and scarcity of ID *image samples*, we leverage CLIP (Radford et al., 2021), connecting text with images, engineering learnable ID and OOD *textual prompts*. we first generate three kinds of adaptive prompts (learnable ID prompts, label-fixed OOD prompts and label-adaptive OOD prompts). Then, we construct the corresponding prototypes, which effectively reduce the neg-

ative impact of the background. 2) As for the second challenge, a prompt-guided OOD detection module is designed to learn an explicit margin between ID and OOD prompts for precise ID-OOD boundary. 3) About the third challenge, we ingeniously introduce two carefully-designed losses to understand the ID image features: (a) To ensure that the label-adaptive OOD prompts have no similar semantics with ID prompts, we design an OOD alignment loss based on label-fixed OOD prompts and label-adaptive OOD prompts. (b) We utilize the weighted cross-entropy loss with ID images, ID prompts and OOD prompts for multi-prompt contrastive learning.

In summary, our main contributions include:

- We target the challenging multi-diversity few-shot OOD detection task, which randomly utilizes a certain number of images with different diversity from each class for training, and conduct OOD detection on the whole testing dataset. Unlike previous works that only learn ID prompts for training, we construct ID and OOD prompts for each class to fully understand the images.

- We propose a novel AMCN for the challenging few-shot OOD detection task. Three carefully-designed

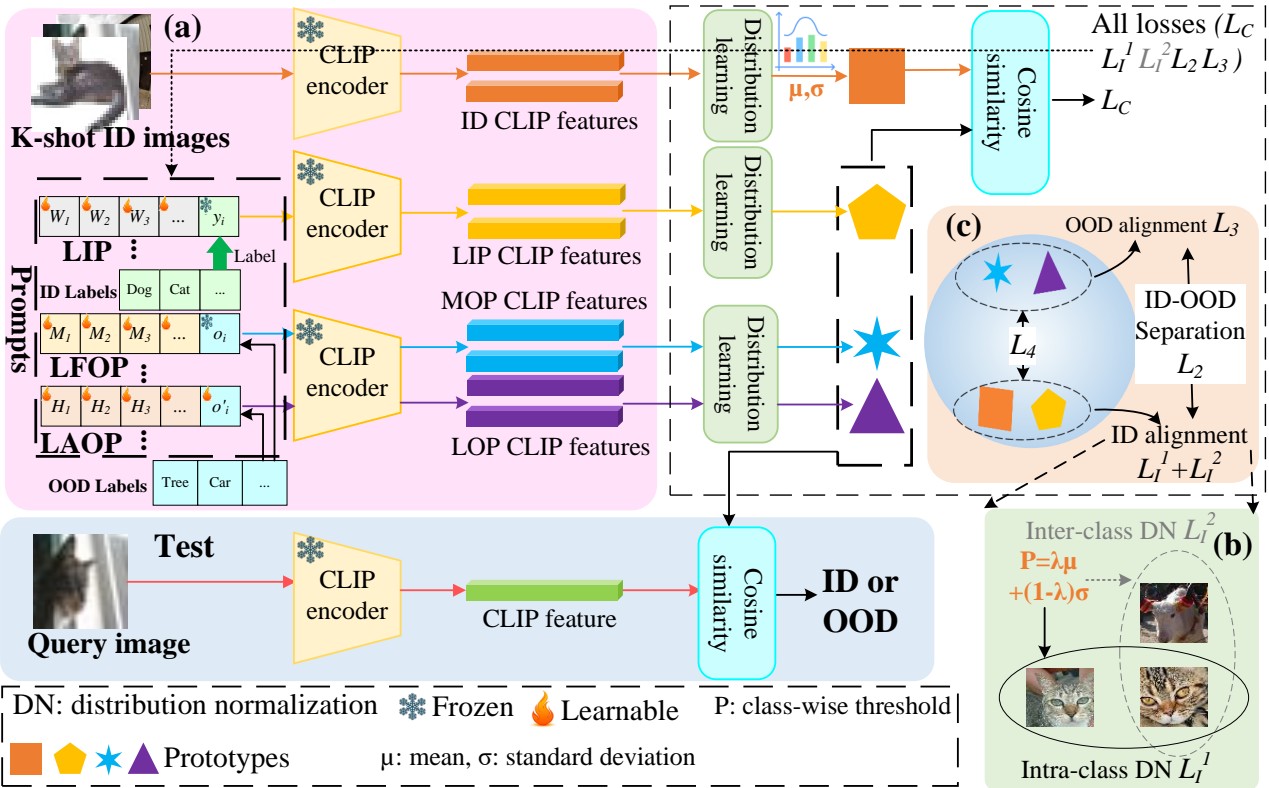

*Figure 2.* Brief framework of our proposed method. (a) denotes the "Adaptive Prompt Generation" module in Section 3.1, (b) denotes the "Prompt-based Multi-diversity Distribution Learning" module in Section 3.2, and (c) denotes the "Prompt-guided OOD Detection" module in Section 3.3. We utilize three module (a,b and c) for training and "Test" for inference. Best viewed in color.

modules are utilized in AMCN to address three challenges.

- Extensive experiments show that our proposed AMCN can significantly outperform existing state-of-the-art works in the few-shot OOD detection task.

## 2. Related Works

### 2.1. Out-of-distribution Detection

As a challenging machine learning task, the out-of-distribution (OOD) detection task aims to detect test samples from distributions that do not overlap with the training distribution. Previous OOD detection methods (Liang et al., 2018b; Liu et al., 2020; Sun et al., 2021; Lee et al., 2018b; Mohseni et al., 2020; Vyas et al., 2018; Yu & Aizawa, 2019; Zaeemzadeh et al., 2021; Hsu et al., 2020; Ming et al., 2023; Jiang et al., 2024) can be divided into four types: classification-based methods (Hendrycks & Gimpel, 2016; Liang et al., 2018b; Lee et al., 2018c;a), density-based methods (Kirichenko et al., 2020; Serrà et al., 2019), distance-based methods (Techapanurak et al., 2020; Lee et al., 2018b) and reconstruction-based methods (Zhou, 2022; Yang et al., 2022). Although previous works have achieved decent suc-

cess, most of them require all the samples for training. Besides, they ignore the different levels of diversity between different classes. Different from these OOD detection methods, we aim at a more challenging task: few-shot OOD detection with multi-diversity distribution.

### 2.2. Prompt Learning

Prompt learning (Gao et al., 2024; Park et al., 2024; Kim et al., 2024) is an emerging area in natural language processing that leverages prompts or instructions to guide pre-trained language models like GPT (Brown et al., 2020), BERT (Devlin, 2018), T5 (Raffel et al., 2020), *etc.*, to perform various downstream tasks. Prompt learning approaches (Liu et al., 2023; Pouramini & Faili, 2024; Xing et al., 2024) aim to bridge the gap between pre-training and fine-tuning by providing models with task-specific context or guidance in the form of natural language prompts. However, most prompt learning works (Pouramini & Faili, 2024; Xing et al., 2024) refer to the closed-set assumption, and cannot be directly utilized into challenging few-shot OOD detection task.

## 2.3. Few-shot Learning

Few-shot learning (Hu et al., 2024; Zhang et al., 2025; Hu et al., 2025; Wang et al., 2025) is a branch of machine learning that focuses on building models capable of learning new concepts with only a few examples. Unlike traditional machine learning models that require large amounts of labeled data to generalize well, few-shot learning aims to make the most out of limited data, mimicking the human ability to learn from only a few examples. Since only a few samples can be used during few-shot training, we often obtain limited knowledge from these training samples. Obviously, our targeted few-shot OOD detection is more challenging than traditional OOD detection.

## 3. Methodology

**Problem definition.** For the $K$-shot OOD detection task, it aims to use only $K$ labeled ID images from each class for model training, and to test on the complete test set for OOD detection. For the training process, we denote the training set as: $D^{id} = \{(x_i, y_i) | i \in \{1, ..., N\}, y_i \in \{1, ..., C\}\}$ with $N$ labeled images from $C$ ID classes. We denote $D^{ood} = (x^{ood}, y^{ood})$ as the OOD dataset, where $x^{ood}$ is the input OOD image, and $y^{ood} \in Y^{ood} := C + 1, ..., C + O$ denotes the OOD label, where $O$ is the OOD label number. Please note that the OOD labels are unknown during training, and they have no overlap of classes with ID labels, *i.e,.*, $Y^{ood} \cap Y^{id} = \phi$. Please note that the OOD data $D^{ood}$ is inaccessible during training.

**Pipeline.** We present our pipeline in Figure 2. Firstly, we utilize the pretrained CLIP encoder (Radford et al., 2021) to extract the image features. Then, we generate adaptive prompts for ID classification. Specifically, we combine $P$ learnable ID prefixes and the label name to generate the learnable ID prompts (LIPs). Also, we generate $S$ label-fixed OOD prompts (LFOPs) by introducing OOD labels from other datasets that disjoint with the ID label set. Since the introduced OOD labels are often limited, we explore $Z$ label-adaptive OOD prompts (LAOPs) for each ID prompt. Besides, we align the image features and ID prompt features by a prompt-guided contrastive loss for ID classification. Moreover, we learn the different distributions of all the classes for adaptive ID alignment. Finally, a prompt-guided OOD detection module is designed to control the explicit margin between ID and OOD prompts for OOD detection.

## 3.1. Adaptive Prompt Generation for ID Classification

In real-world applications, we only access ID samples during training. Therefore, it is unrealistic to directly obtain the OOD prompt for the future OOD detection task. Given a sentence, we can obtain different sentences with various semantics by changing the prefix of the sentence or replacing

the class label. For the text prompt for class $y_i$, we follow the popular predefined templates: "a photo of a $[y_i]$", where $[y_i]$ denotes the corresponding class name. Thus, we can design the learnable ID prompt as follows:

$$f_{lip}^i = [W_1][W_2] \dots [W_{N_{IP}}][y_i], \qquad (1)$$

where $N_{IP}$ denotes the length of the ID prefix and $[W_i]$ denotes the $i$-th text token learned from the CLIP network. By Eq. (1), we can construct the learnable ID prompt.

Similarly, we generate OOD prompts from the OOD labels in other large-scale datasets, which can provide partial knowledge about OOD samples (Yang et al., 2024; Liu et al., 2021; Cao et al., 2024). In real-world applications, we can obtain partial knowledge about OOD samples. For example, when we treat the CIFAR-10 dataset (Krizhevsky, 2009) as ID, we can use "chair" in the CIFAR-100 dataset (Krizhevsky, 2009) as OOD since CIFAR-10 has the "chair" class. In this way, we want to generate two types of label-adaptive OOD prompts: label-fixed OOD prompts and label-adaptive OOD prompts. The label-fixed OOD prompt will introduce some human knowledge to assist our model for OOD detection. As for the label-adaptive OOD prompt, we let it learn prefixes and labels by itself. To obtain the OOD prompt based on the ID prompt and the OOD labels, we utilize a similar process to generate the adaptive OOD prompts:

$$
\begin{aligned}
f_{lfop}^i &= [M_1] \dots [M_{N_{lfop}}][o_i], \\
f_{laop}^i &= [H_1] \dots [H_{N_{laop}}][o_i'],
\end{aligned} \qquad (2)
$$

where $[o_i]$ is the OOD label from other datasets ($D^{ood}$) that disjoint with $D^{id}$; $[o_i']$ is the learnable label, which is initialized by $[o_i]$; $N_{laop}$ denotes the length of label-adaptive OOD prefix, $f_{lfop}^i$ and $f_{laop}^i$ denote the label-fixed OOD prompt and label-adaptive OOD prompt, respectively. For $f_{lfop}^i$, its prefix is learnable and its label is fixed. As for $f_{laop}^i$, both its prefix and its label are learnable. Based on $f_{lfop}^i$, we can handle various prefix structures; by $f_{laop}^i$, we are able to explore different latent OOD labels. To ensure that the prompt feature dimensions are consistent, we make all the text encoders share the parameter weights. Besides, we align ID prompts with corresponding ID images, and push OOD prompts away from ID images. During training, since more negative prompts (OOD prompts) will help us conduct better multi-prompt contrastive learning for guidance, we utilize all OOD prompts to compare with ID images. For convenience, we introduce a similarity function $\mathcal{S}(a, b)$ for two inputs ($a$ and $b$) as follows:

$$\mathcal{S}(a, b) = \exp[1/\sigma \cdot \cos(a, b)], \qquad (3)$$

where $\cos(\cdot, \cdot)$ denotes the cosine similarity; $\sigma$ is a temperature parameter. Therefore, we introduce the following

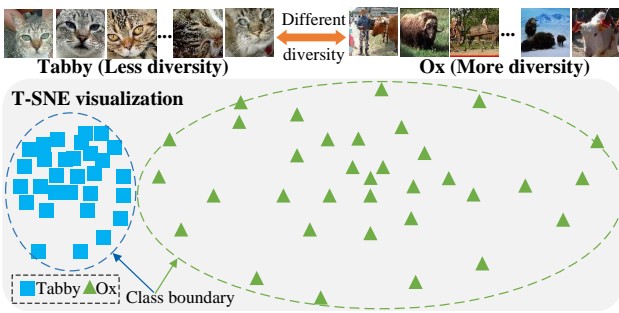

**Tabby (Less diversity)** · · · **Ox (More diversity)**

T-SNE visualization

Tabby  Ox  Class boundary

*Figure 3.* T-SNE visualization of diversity for different classes on ImageNet-1k.

prompt learning loss for ID classification:

$$\mathcal{L}_C = \mathbb{E}_{f_x^i}\left[-\log \frac{\tau_1 \sum_{f_x^i} \mathcal{S}(f_x^i, \bar{f}_{lip}^i)}{\tau_1 \sum_{f_x^i} \mathcal{S}(f_x^i, \bar{f}_{lip}^i) + (1-\tau_1)\sum_{f_{op}^i \in F_o} \mathcal{S}(f_x^i, f_{op}^i)}\right],$$ (4)

where $f_x^i$ denotes ID image feature; $\bar{f}_{lip}^i = ave(f_{lip}^1, f_{lip}^2, ..., f_{lip}^P)$ is the prototype of ID prompt features for the $i$-th image, where $ave(\cdot)$ denotes the average pooling; $F_o = \{ave(f_{op}^i)|f_{op}^i \in \{f_{lfop}^i\}_{i=1}^{N_{lfop}} \cup \{f_{laop}^i\}_{i=1}^{N_{laop}}\}$ is a set of OOD prompt features. Based on $\mathcal{L}_1$, we can train our ID classifier by ID prompts and OOD prompts.

### 3.2. Prompt-based Multi-diversity Distribution Learning for Adaptive ID Alignment

In real-world applications, different classes indeed exhibit varying degrees of sample diversity. This diversity, which reflects how varied the images within a class are, can be influenced by multiple factors, including the nature of the class, its semantic breadth, and the challenges of collecting representative samples.

In fact, there is a distribution gap between unseen ID samples (*i.e.*, not selected in $K$ samples) and OOD samples. Since these unseen ID samples is not used for training, previous OOD detection methods might treat these unseen ID samples as OOD samples, which will lead to incorrect detection results. In addition, these methods utilize the same threshold for all classes, seriously limiting their performance in real-world complex applications. Motivated by the effectiveness of normal distribution, we try to learn the threshold for each ID class.

For any class $c$, we name these samples with $y \neq c$ as pseudo-OOD samples, where $c \in \{1, ..., C\}$ is the corresponding class of $f_x^i$. If a sample is the pseudo-OOD sample for all the classes, it is a real OOD sample. For a dataset with 2 classes ("cat" and "dog"), a "dog" sample is the pseudo-OOD sample for the "cat" class, while a "tree"

sample is the real OOD sample for both "cat" and "dog" classes. In this section, we design an adaptive distribution extraction module to fully learn the distribution of each class based on only a few samples and fine-tune the prediction output of any test sample.

**Learning distribution.** Since the mean and the standard deviation are two significant metrics to learn the distribution, we calculate them in each class. Given $K$ ID training samples $\{x_i\}_{i=1}^K \in D^{id}$ in the $c$-th class, we estimate the mean $\mu_{in}$ and standard deviation $\sigma_{in}$ as follows:

$$\mu_c = \frac{\sum_{i=1}^K \mathbb{S}_c(x_i)}{K},$$

$$\sigma_c = \sqrt{\frac{\sum_{i=1}^K (\mathbb{S}_c(x_i) - \mu_c)^2}{K-1}},$$ (5)

where $\mathbb{S}_c(x_i)$ is a class distribution score, which is defined as follows:

$$\mathbb{S}_c(x_i) = \frac{\exp(o_c(x_i))}{\tau_0 + \mathcal{M}_c^{pse}},$$ (6)

where $\tau_0$ is a parameter adjustable as need, $o_c(x_i)$ denotes the logit output of sample $x_i$ in class $c$, and $\mathcal{M}_c^{pse} \in \mathbb{R}$ denotes the initial pseudo-OOD distribution. To adapt the pseudo-OOD distribution $\mathcal{M}_c^{pse}$, we first use the OOD filter to predict OOD samples, and then conduct a momentum update of $\mathcal{M}_c^{pse}$ during inference. Based on the above process, $\mathcal{M}_c^{pse}$ is updated as the mean of distribution for the predicted OOD samples. For the pseudo-OOD distribution $\mathcal{M}_c^{pse}$, we first initialize its entries based on the mean distribution of the pseudo-OOD samples. Then, we update these entries by an online fashion module.

As shown in Figure 3, different classes have distinct diversity. Therefore, a diversity-guided decision boundary is required for each class during classification. We propose the following novel P-score as the class-wise threshold for diversity-guided decision boundary:

$$P_c = \lambda \cdot \mu_c + (1-\lambda) \cdot \sigma_c,$$ (7)

where $\lambda$ is a parameter to balance the weight of mean and standard deviation. Based on $\mathbb{S}_c(x_i)$ and $P_c$, we can obtain:

$$x_i \text{ belongs to } \begin{cases} \text{pseudo-OOD}, & \mathbb{S}_c(x_i) > P_c, \\ \text{class } c, & \mathbb{S}_c(x_i) \leq P_c. \end{cases}$$ (8)

Based on (8), if a sample $x_i$ is pseudo-OOD for all the classes, it will be detected as real OOD. If any sample $x_i$ is detected as pseudo-OOD sample, we can update the distribution $\mathcal{M}_c^{pse}$ as follows:

$$\mathcal{M}_c^{pse}(t) = \begin{cases} \mathcal{M}_c^{pse}(t-1), & \mathbb{S}_c(x_i) > P_c, \\ \frac{\exp(o_c(x_i)) + O \cdot \mathcal{M}_c^{pse}(t-1)}{O+1}, & \mathbb{S}_c(x_i) \leq P_c, \end{cases}$$ (9)

where $t$ denotes the $t$-th iteration during training and $O$ denotes the number of predicted OOD samples. We only keep $O$ and current $\mathcal{M}_c^{pse}$ unchanged during inference.

Previous OOD detection works utilize common global distribution loss to learn the vanilla pseudo-OOD distribution $\mathcal{M}_c^{pse}$. Besides, we have to manually fine-tune the sensitive hyperparameters on distribution margins in complex datasets under the challenging few-shot setting, which might result in an sensitive OOD filter and then destroy the distribution learning. To this end, we aim to explore intra-class distribution and inter-class distribution.

**Intra-class distribution normalization.** To fully learn the intra-class distribution of ID samples for better classification, we independently normalize the distribution for each class by the following loss:

$$\mathcal{L}_I^1 = \sum_{c=1}^{C} (\mathbb{E}_{(f_x^i, c)}[(\max(0, \epsilon_1 - \sum_{i=1}^{B} \frac{\tau_1 \sum_{f_x^i} \mathcal{S}(f_x^i, \bar{f}_{lip}^i)}{M_i}))^2]$$
$$+ \mathbb{E}_{f_{op}^i \in F_o}[(\max(0, \sum_{i=1}^{B} \frac{(1-\tau_1)\sum_{f_{op}^i \in F_o} \mathcal{S}(f_x^i, f_{op}^i)}{M_i} - \epsilon_2))^2]),$$
(10)

where $\epsilon_1$ and $\epsilon_2$ are two hyper-parameters, $B$ denotes the batch size, and $M_i$ is defined as:

$$M_i = \tau_1 \sum_{f_x^i} \mathcal{S}(f_x^i, \bar{f}_{lip}^i) + (1-\tau_1)\sum_{f_{op}^i \in F_o} \mathcal{S}(f_x^i, f_{op}^i), \quad (11)$$

where $\tau_1$ is a parameter adjustable as need, Based on $\mathcal{L}_I^1$, we can balance the the sum of distribution score on all ID samples for each ID class within a batch.

**Inter-class distribution normalization.** Similar to intra-class distribution normalization, we balance the distributions of all the classes by the following loss:

$$\mathcal{L}_I^2 = \mathbb{E}_{(f_x^i, c)}[(\max(0, \epsilon_3 - \sum_{c=1}^{C} \frac{\tau_1 \sum_{f_x^i} \mathcal{S}(f_x^i, \bar{f}_{lip}^i)}{M_i}))^2]$$
(12)
$$+ \mathbb{E}_{f_{op}^i \in F_o}[(\max(0, \sum_{c=1}^{C} \frac{(1-\tau_1)\sum_{f_{op}^i \in F_o} \mathcal{S}(f_x^i, f_{op}^i)}{M_i} - \epsilon_4))^2],$$

where $\epsilon_3$ and $\epsilon_4$ are two hyper-parameters.

By integrating $\mathcal{L}_C$, $\mathcal{L}_I^1$ and $\mathcal{L}_I^2$, we can obtain the final ID classification loss as follows:

$$\mathcal{L}_1 = \mathcal{L}_C + \mathcal{L}_I^1 + \mathcal{L}_I^2. \quad (13)$$

### 3.3. Prompt-guided OOD Detection

The realistic OOD detection networks always face the following challenges: 1) we rarely obtain OOD images. 2) The OOD prompts (label-fixed and label-adaptive) in Eq. (4) are treated equally, and only treat ID image features to generate negative samples for multi-prompt contrastive learning, which might lead to an unclear margin between ID and OOD prompt and wrong OOD detection results. We

observe that many OOD prompts can help us understand the ID images. For example, "a photo of a cat" contains the ID semantics (cat), and we can change the label to obtain an OOD prompt "a photo of a chair", which corresponds to the OOD semantics. To update these OOD prompts, we initialize their embeddings and introduce a weighted OOD alignment loss. Then, we integrate ID prompts and OOD prompts by a multi-prompt contrastive learning strategy for OOD detection.

*Remark* 3.1. In our proposed AMCN, all final features are projected onto the unit hyper-sphere for cross-modal matching.

Therefore, we propose a novel prompt-guided ID-OOD separation module to generate an explicit margin between ID and OOD prompt features. Thus, we introduce the following prompt-guided ID-OOD separation loss:

$$\mathcal{L}_2 = \mathbb{E}_{f_x^i}\left[-\min\left(0, e(\frac{f_x^i}{\|f_x^i\|_2}, \frac{\bar{f}_{op}^i}{\|\bar{f}_{op}^i\|_2}) - e(\frac{f_x^i}{\|f_x^i\|_2}, \frac{\bar{f}_{lip}^i}{\|\bar{f}_{lip}^i\|_2})\right)\right],$$
(14)

where $e(\cdot, \cdot)$ denotes the euclidean distance. To mine the latent OOD information, we conduct weighted average on all OOD prompt features to generate the final OOD prototype $\bar{f}_{op}^i$:

$$\bar{f}_{op}^i = \frac{1}{S+Z}[\sum_{i=1}^{S} ave(f_{lfop}^i) + \sum_{i=1}^{Z} ave(f_{laop}^i)]. \quad (15)$$

Similarly, we normalize the features in $\mathcal{L}_2$ and set the margin to 0. Unlike $\mathcal{L}_1$, $\mathcal{L}_2$ attempts to generate a larger margin between ID samples and the OOD prototype than between ID samples and the ID prototype, allowing our model to correctly distinguish ID and OOD prototypes. To keep the label-adaptive OOD prompts away from the ID prompts, we conduct the following weighted OOD alignment based on label-fixed OOD prompts and label-adaptive OOD prompts:

$$\mathcal{L}_3 = \sum_i \left\| \frac{\tau_2 \cdot \bar{f}_{laop}^i}{\|\bar{f}_{laop}^i\|_2} - \frac{(1-\tau_2) \cdot \bar{f}_{lfop}^i}{\|\bar{f}_{lfop}^i\|_2} \right\|_2^2, \quad (16)$$

where $\|\cdot\|_2$ is the L2-norm; $\tau_2$ is a weight parameter; for the $i$-th image, $\bar{f}_{lfop}^i$ and $\bar{f}_{laop}^i$ are respectively the feature prototypes of label-fixed OOD prompts and label-adaptive OOD prompts.

Similar to the ID classification loss $\mathcal{L}_1$, we design the multi-prompt contrastive learning loss for OOD detection:

$$\mathcal{L}_4 = \mathbb{E}_{\bar{f}_{op}^i}\left\{-\log \frac{\tau_3 \sum_{\bar{f}_{op}^i} \mathcal{S}(f_x^i, \bar{f}_{op}^i)}{(1-\tau_3)\sum_{\bar{f}_{lip}^i} \mathcal{S}(f_x^i, \bar{f}_{lip}^i) + \tau_3 \sum_{\bar{f}_{op}^i} \mathcal{S}(f_x^i, \bar{f}_{op}^i)}\right\},$$
(17)

where $\tau_3$ is a weight parameter. Based on $\mathcal{L}_4$, we can minimize the similarity between ID prompts and OOD prompts for clearer ID-OOD separation.

The overall loss function with balanced hyperparameters $(\alpha_1, \alpha_2$ and $\alpha_3)$ for training is as follows:

$$\mathcal{L} = \mathcal{L}_1 + \alpha_1\mathcal{L}_2 + \alpha_2\mathcal{L}_3 + \alpha_3\mathcal{L}_4, \quad (18)$$

where $\alpha_1$, $\alpha_2$ and $\alpha_3$ are parameters to balance the significance between different losses.

# 4. Experiments

## 4.1. Experimental Setup

**Datasets.** For fair comparison, we follow MOS (Huang & Li, 2021) and MCM (Ming et al., 2022) to utilize ImageNet-1k (Deng et al., 2009) as ID set and a subset of iNaturalist (Horn et al., 2018), PLACES (Zhou et al., 2018), TEXTURE (Cimpoi et al., 2014) and SUN (Xiao et al., 2010) as OOD set. For each OOD set, the classes are not overlapping with the ID set. Also, we follow (Huang & Li, 2021) to randomly select these OOD data from the classes disjointing from ImageNet-1k (Deng et al., 2009). Please refer to (Huang & Li, 2021; Miyai et al., 2023; Ming et al., 2022) for more dataset details and public dataset split.

**Implementation details.** Following (Miyai et al., 2023), we adopt CLIP-ViT-B/16 (Radford et al., 2021) as the pre-trained model for OOD prompt learning. For the few-shot setting (Ye et al., 2020), following previous works (Miyai et al., 2023; Ye et al., 2020), we try different shots (1, 2, 4, 8, 16). For the parameters, we set $\alpha_1 = 0.4, \alpha_2 = 0.2, \alpha_3 = 0.8, \theta = 0.8, \tau = 1.0, \gamma = 0.7, P = 1, S = 50, Z = 50$. We set AdamW (Loshchilov & Hutter, 2019) as the optimizer, the learning rate of 0.003, the batch size as 64, the token length as 16 and the training epoch as 100. Codes are available in Github.

**Evaluation metrics.** Following (Miyai et al., 2023; Bukhsh & Saeed, 2023; Nakamura et al., 2024; Kahya et al., 2024), we employ three popular evaluation metrics: FPR95, AUROC and ACC. FPR95 means the false positive rate of OOD samples when the true positive rate of ID samples is at 95%. AUROC measures the area under the receiver operating characteristic curve.

## 4.2. Comparison With State-of-the-arts

**Compared methods.** To comprehensively analyze the performance of our model, we compare our model with four types of state-of-the-art OOD detection models: fully-supervised (Full), zero-shot, one-shot and eight-shot, where fully-supervised methods and zero-shot methods are baselines for performance comparison. The following open-source methods are selected for performance comparison. 1) Fully-supervised: ODIN (Liang et al., 2018a), ViM (Wang et al., 2022), KNN (Sun et al., 2022), NPOS (Tao et al., 2023). 2) Zero-shot: MCM (Ming et al., 2022), SeTAR (Li et al., 2024b) with MCM Score and GL-MCM (Miyai et al., 2025). 3) One-shot and eight-shot: CoOp (Zhou et al., 2022) and LoCoOp (Miyai et al., 2023), SCT (Yu et al., 2024).

**Experimental results.** As shown in Table 1 and Figure

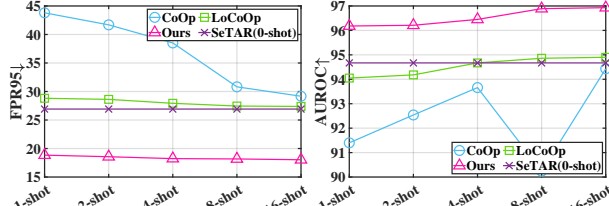

*Figure 4.* Performance of different shots on iNaturalist.

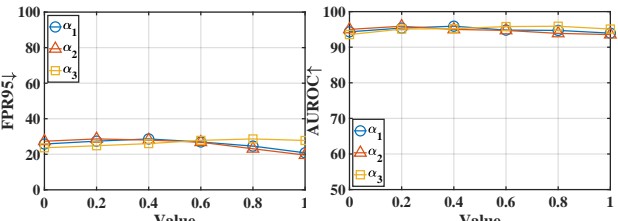

*Figure 5.* Parameter analysis on SUN.

4, we compare our proposed method with state-of-the-art methods, where our proposed method achieves the best performance in all the cases. In particular, with the Texture dataset as the OOD set in terms of "FPR95", our method outperforms state-of-the-art method SCT by 9.71% under the one-shot setting. The significant performance improvement is mainly because our method can first construct adaptive ID and OOD prompts by only limited labeled ID images, and then conduct prompt-based ID-OOD separation. On the SUN dataset, compared with SCT in terms of "AUROC", our method improves the OOD detection performance by 0.27% under the one-shot setting and by 1.12% under the eight-shot setting. The main reason is that our method can effectively learn the distribution of each class to reduce the negative impact of multi-diversity distribution on SUN. In Figure 4, we compare two representative few-shot OOD detection works (LoCoOp and CoOp) under different few-shot settings. Obviously, our method outperforms LoCoOp and CoOp in all the cases by a large margin. In many cases, LoCoOp and CoOp even perform worse than zero-shot GL-MCM, while our method significantly outperforms GL-MCM. The core reason is that LoCoOp and CoOp are misled by the various background information in the images. Different from LoCoOp and CoOp, we can learn the proper prototypes for each input (ID image, learnable ID prompt, label-fixed OOD prompt and label-adaptive OOD prompt) based on the limited images to handle various backgrounds.

**Visualization results.** To qualitatively investigate the effectiveness of our method, we report a representative example. As shown in Figure 6, our method achieves better performance in both ID classification and OOD detection, which further shows the effectiveness of our method.

*Table 1.* Performance comparison for the few-shot OOD detection task. We direct cite the results of compared methods from corresponding works.

| OOD set | | Texture | | Places | | SUN | | iNaturalist | | Average | |
|---|---|---|---|---|---|---|---|---|---|---|---|
| Method | Shot | FPR95↓ | AUROC↑ | FPR95↓ | AUROC↑ | FPR95↓ | AUROC↑ | FPR95↓ | AUROC↑ | FPR95↓ | AUROC↑ |
| KNN | Full | 64.35 | 85.67 | 39.61 | 91.02 | 35.62 | 92.67 | 29.17 | 94.52 | 42.19 | 90.97 |
| ViM | Full | 53.94 | 87.18 | 60.67 | 83.75 | 54.01 | 87.19 | 32.19 | 93.16 | 50.20 | 87.82 |
| ODIN | Full | 51.67 | 87.85 | 55.06 | 85.54 | 54.04 | 87.17 | 30.22 | 94.65 | 47.75 | 88.80 |
| NPOS | Full | 46.12 | 88.80 | 45.27 | 89.44 | 43.77 | 90.44 | 16.58 | 96.19 | 37.94 | 91.22 |
| GL-MCM | 0 | 57.93 | 83.63 | 38.85 | 89.90 | 30.42 | 93.09 | 15.16 | 96.71 | 35.59 | 90.83 |
| MCM | 0 | 57.77 | 86.11 | 44.69 | 89.77 | 37.67 | 92.56 | 31.86 | 94.17 | 43.00 | 90.65 |
| SeTAR | 0 | 55.83 | 86.58 | 42.64 | 90.16 | 35.57 | 92.79 | 26.92 | 94.67 | 40.24 | 91.05 |
| CoOp | 1 | 50.64 | 87.83 | 46.68 | 89.09 | 38.53 | 91.95 | 43.80 | 91.40 | 44.91 | 90.07 |
| LoCoOp | 1 | 49.25 | 89.13 | 39.23 | 91.07 | 33.27 | 93.67 | 28.81 | 94.05 | 37.64 | 91.98 |
| SCT | 1 | 48.87 | 86.66 | 32.81 | 91.23 | 23.52 | 94.58 | 19.16 | 95.70 | 31.09 | 92.04 |
| **Ours** | **1** | **39.16** | **89.88** | **32.76** | **92.78** | **23.26** | **94.85** | **18.84** | **96.18** | **30.87** | **92.47** |
| CoOp | 8 | 43.29 | 89.92 | 41.17 | 89.76 | 34.45 | 92.50 | 38.52 | 90.24 | 39.36 | 90.61 |
| LoCoOp | 8 | 42.49 | 90.98 | 40.53 | 91.53 | 33.87 | 93.23 | 27.45 | 94.86 | 36.09 | 92.40 |
| SCT | 8 | 40.35 | 91.82 | 38.77 | 92.41 | 23.48 | 94.77 | 18.65 | 95.82 | 32.32 | 93.53 |
| **Ours** | **8** | **38.31** | **93.43** | **32.45** | **93.96** | **23.17** | **95.89** | **18.17** | **96.89** | **30.56** | **94.29** |

*Table 2.* Main ablation study on Texture, where "M1" is "Adaptive Prompt Generation", "M2" is "Prompt-based Multi-diversity Distribution Learning" and "M3" is "Prompt-guided OOD Detection". We remove each key individual component and keep the other two modules to investigate its effectiveness.

| M1 | M2 | M3 | One-shot | | Eight-shot | |
|---|---|---|---|---|---|---|
| | | | FPR95↓ | AUROC↑ | FPR95↓ | AUROC↑ |
| ✗ | ✓ | ✓ | 41.36 | 82.59 | 40.82 | 86.24 |
| ✓ | ✗ | ✓ | 40.95 | 83.24 | 40.26 | 86.88 |
| ✓ | ✓ | ✗ | 40.35 | 83.19 | 39.72 | 87.92 |
| ✓ | ✓ | ✓ | **39.16** | **89.88** | **38.31** | **93.43** |

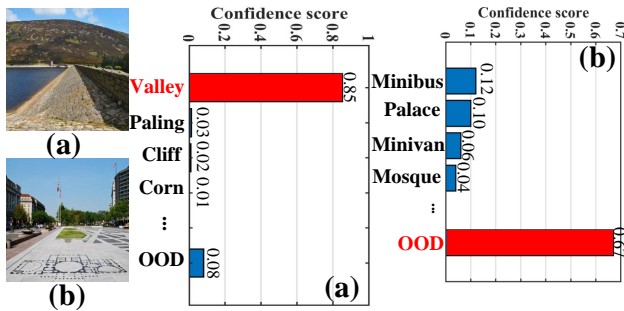

*Figure 6.* Visualization results for (a) ID classification and (b) OOD detection on Places.

*Table 3.* Ablation study about different prompts on Texture, where we remove each prompt to investigate its effectiveness.

| LIP | LFOP | LAOP | One-shot | | Eight-shot | |
|---|---|---|---|---|---|---|
| | | | FPR95↓ | AUROC↑ | FPR95↓ | AUROC↑ |
| ✗ | ✓ | ✓ | 40.27 | 86.23 | 38.87 | 89.01 |
| ✓ | ✗ | ✓ | 40.52 | 85.67 | 39.36 | 88.37 |
| ✓ | ✓ | ✗ | 39.80 | 87.59 | 39.15 | 89.50 |
| ✓ | ✓ | ✓ | **39.16** | **89.88** | **38.31** | **93.43** |

## 4.3. Ablation Study

**Main ablation study.** To demonstrate the effectiveness of each module in our model, we conduct ablation studies on Texture in Table 2. Based on Table 2, we can observe that all three modules contribute a lot to the final performances under both the one-shot setting and the eight-shot setting, which shows the effectiveness of our well-designed prompts

for the challenging few-shot OOD detection task. As the core module, "Adaptive Prompt Generation" can generate adaptive prompts to fully understand the images and labels by three adaptive prompts (LIP, LFOP and LAOP) for ID classification. In Texture, different classes have distinct diversities, which makes many few-shot OOD detection methods difficult to learn the correct distribution for each class. Fortunately, "Prompt-based Multi-diversity Distribution Learning" can effectively learn the intra-class distribution and inter-class distribution, which reduces the negative impact of distinct diversity of different classes. Based on three adaptive prompts, "Prompt-guided OOD detection" can correctly detect OOD samples in Texture.

**Importance of different prompts.** In the "Adaptive Prompt Generation" module, we design three adaptive prompts (LIP, LFOP and LAOP) for each labeled ID sample. To show the importance of different prompts, we conduct an ablation

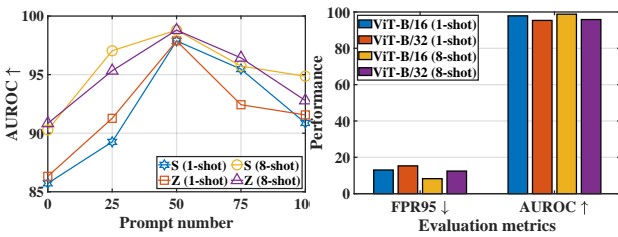

*Figure 7.* Ablation study on iNaturalist, where the left figure is the ablation about the prompt number and the right figure is the ablation about feature encoders.

*Table 4.* Ablation study on adaptive threshold on iNaturalist.

| Threshold | One-shot | | Eight-shot | |
|---|---|---|---|---|
| type | FPR95↓ | AUROC↑ | FPR95↓ | AUROC↑ |
| Fixed | 20.70 | 93.83 | 20.51 | 94.16 |
| **Adaptive** | **18.84** | **96.18** | **18.17** | **96.89** |

study on different prompts in Table 3. All three prompts can bring a significant performance improvement under different settings, showing the effectiveness of our three prompts in the challenging few-shot OOD detection task.

**Effect of the prompt number.** We further conduct an ablation study to analyze the impact of the negative prompt numbers (LFOP number $S$ and LAOP number $Z$) on iNaturalist. As shown in Figure 7, we can observe that, with the increase of $K$, the variation of the performance follows a general trend, *i.e.*, rises at first and then starts to decline. The optimal LFOP number $S$ is 50 and the optimal LAOP number $Z$ is 50. Thus, we set $S = Z = 50$ in this paper.

**Influence of adaptive threshold.** In our "Prompt-based Multi-diversity Distribution Learning" module, we design an adaptive threshold $P_c$ for each class. To assess the performance of the adaptive threshold $P_c$, we change the threshold to obtain the ablation models. Table 4 illustrates the performance comparison for different models. Obviously, our full model obtains the best results since our module can generate an adaptive threshold, which illustrates the effectiveness of our adaptive threshold to learn both inter-class distribution and inter-class distribution on multi-diversity dataset iNaturalist.

**Parameter analysis.** We introduce some losses to supervise the training process. To evaluate the significance of these losses, we conduct experiments on the SUN dataset under the eight-shot setting. Figure 5 presents the ablation study on the hyper-parameters $(\alpha_1, \alpha_2, \alpha_3)$. We can find that, their performance only varies in a small range, indicating that our model is insensitive to these parameters. Our model can achieve the best performance with $\alpha_1 = 0.4, \alpha_2 = 0.2, \alpha_3 = 0.8$. Therefore, we utilize the above parameter setting in our all experiments.

## 5. Conclusion

In this paper, we target a challenging machine learning task: few-shot OOD detection, which only uses a few labeled ID images from each class to train the designed model and to test the complete test set for OOD detection. To address it, we propose a novel method, AMCN, which generates adaptive prompts by the given label set to learn the distribution of each class for adaptive OOD detection. Experimental results on multiple challenging benchmarks demonstrate the effectiveness of our proposed method.

## Acknowledgement

This research was conducted as part of the DesCartes program and was supported by the National Research Foundation, Prime Minister's Office, Singapore, under the Campus for Research Excellence and Technological Enterprise (CREATE) program. This research is also supported by the National Research Foundation, Singapore and DSO National Laboratories under the AI Singapore Programme (AISG Award No: AISG2-RP-2020-017). The computational work for this article was (fully/partially) performed on resources of the National Supercomputing Centre, Singapore (https://www.nscc.sg).

## Impact Statement

This paper presents work whose goal is to advance the field of Machine Learning. There are many potential societal consequences of our work, none which we feel must be specifically highlighted here.

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

## A. More Details

*Remark* A.1. In our proposed AMCN, all final features are projected onto the unit hyper-sphere for cross-modal matching.

*Proof.* 1) Embedding computation: In our framework, we use a dual encoder architecture where an image encoder $f_{image}(x)$ and a text encoder $f_{text}(y)$ map images and texts into a shared feature space. The image and text encoders generate feature vectors $z_{image} = f_{image}(x)$ and $z_{text} = f_{text}(y)$, respectively.

2) Feature normalization: After encoding, we apply L2 normalization to both image and text feature vectors. L2 normalization scales a vector $z$ to have unit norm:

$$\hat{z} = \frac{z}{||z||_2} \tag{19}$$

This normalization step ensures that all features lie on the surface of the unit hyper-sphere in the feature space.

3) Multi-modal contrastive loss mechanism: The contrastive loss is based on cosine similarity:

$$\cos(z_{image}, z_{text}) = \frac{z_{image} \cdot z_{text}}{||z_{image}||_2 \cdot ||z_{text}||_2} \tag{20}$$

Because the features $\hat{z}_{image}$ and $\hat{z}_{text}$ are normalized to unit norm, this reduces to the dot product:

$$\cos(\hat{z}_{image}, \hat{z}_{text}) = \hat{z}_{image} \cdot \hat{z}_{text}. \tag{21}$$

This formulation requires and enforces the projection of all feature vectors onto the unit hyper-sphere.

4) Unit Hyper-Sphere Condition: By the definition of L2 normalization, for any feature vector $z$, we have:

$$||\hat{z}||_2 = 1. \tag{22}$$

This property confirms that the features are constrained to the unit hyper-sphere. Therefore, the use of explicit L2 normalization in our architecture ensures that all features $\hat{z}_{image}$ and $\hat{z}_{text}$ lie on the unit hyper-sphere. □

## B. Discussion about the diversity of different classes

In the ImageNet-1K dataset, different classes indeed exhibit varying degrees of sample diversity. This diversity, which reflects how varied the images within a class are, can be influenced by multiple factors, including the nature of the class, its semantic breadth, and the challenges of collecting representative samples. Here's a breakdown of why and how this occurs:

**1) Semantic breadth of classes.** Fine-grained classes (e.g., "Persian cat" vs. "Siamese cat") have low sample diversity because they represent very specific objects or subtypes. Images in these classes tend to have high visual similarity. Broad classes (e.g., "dog" or "tree") encompass a wide range of subtypes and contexts, leading to high sample diversity. Example: The "dog" class might include different breeds, poses, environments, and lighting conditions, while "toaster" images primarily focus on the object itself with fewer variations.

**2) Intrinsic variability of the object.** Some objects inherently have more variability. For example, "clouds" can appear in countless shapes, colors, and settings, while "keyboard" is typically constrained to a small range of appearances and layouts. classes like "person" exhibit enormous diversity due to differences in age, ethnicity, clothing, posture, and activities.

**3) Contextual variability.** Classes that often appear in diverse contexts, such as "car" (urban streets, rural areas, different weather conditions), have higher diversity compared to objects like "microwave", which is typically photographed indoors in controlled settings.

**4) Collection bias.** The way data is collected can introduce bias and affect diversity. For instance: Popular classes might include diverse images due to abundant online resources. Rare or niche classes may be underrepresented, with fewer images and less diversity. Example: Images of "golden retrievers" might be sourced from both professional and amateur photography, increasing diversity. Meanwhile, classes like "goldfish bowl" might rely on a limited set of typical scenes.

**5) Impact of ImageNet design choices.** ImageNet was curated to balance the number of images per class (usually 1,000 images per class). However, this balancing doesn't guarantee uniform diversity. The emphasis on representativeness might lead to classes with limited diversity being artificially padded with near-duplicate images or slight variations.

**6) Challenges in diverse classes.** High-diversity classes pose challenges for machine learning models, as they must generalize across a wide range of appearances and contexts. Conversely, low-diversity classes might lead to models that perform well on training data but lack robustness to real-world variations.

