# OpenReview forum: "Adaptive Multi-prompt Contrastive Network for Few-shot Out-of-distribution Detection"
_ICML.cc/2025/Conference — ICML 2025 spotlightposter_

### Official Review · Reviewer_oFtr · 2025-03-03

**Overall Recommendation:** 3

**Summary:**

This paper proposes the Adaptive Multi-prompt Contrastive Network for few-shot out-of-distribution detection, aiming to improve OOD detection performance when only limited labeled in-distribution samples are available. The method introduces adaptive prompts to learn class distributions and enhance the separation between ID and OOD samples. It includes three main modules: Adaptive Prompt Generation, Prompt-based Multi-diversity Distribution Learning, and Prompt-guided OOD Detection. These modules address challenges such as background bias and diverse class distributions.

**Claims And Evidence:**

The claims made in the submission are generally well-supported by clear and convincing evidence. The authors propose the Adaptive Multi-prompt Contrastive Network to address the challenges of few-shot out-of-distribution detection and claim that it outperforms existing methods. This claim is substantiated through extensive experiments on multiple benchmark datasets, including ImageNet-1k as the in-distribution dataset and various OOD datasets such as iNaturalist, Places, TEXTURE, and SUN. The results demonstrate significant improvements in key metrics like FPR95 and AUROC across different few-shot settings, with notable reductions in false positive rates and enhanced detection accuracy.

**Essential References Not Discussed:**

There are no essential related works missing from the citations.

**Ethical Review Flag:**

Flag this paper for an ethics review.

**Ethics Expertise Needed:**

["Other expertise"]

**Experimental Designs Or Analyses:**

The authors conducted extensive experiments across multiple benchmark datasets, including ImageNet-1k as the in-distribution dataset and various OOD datasets such as iNaturalist, Places, TEXTURE, and SUN. These datasets are widely recognized in the field and provide a diverse range of scenarios to test the robustness of the proposed method.

The experimental setup includes comparisons with several state-of-the-art methods, covering fully supervised, zero-shot, one-shot, and eight-shot settings. This comprehensive comparison allows for a clear evaluation of AMCN's performance relative to existing approaches under different conditions. The authors also provided detailed ablation studies to analyze the contributions of individual components of their framework, such as different prompt types (LIP, LFOP, LAOP) and the impact of adaptive thresholding.

**Methods And Evaluation Criteria:**

The authors introduce the Adaptive Multi-prompt Contrastive Network, which incorporates adaptive prompts and a multi-prompt contrastive learning framework to address the challenges of limited labeled in-distribution samples and the absence of OOD samples during training. This approach is innovative and tailored to the specific difficulties of few-shot OOD detection, such as the need to generalize from limited data and handle diverse class distributions. The evaluation criteria used in the paper, including FPR95, AUROC, and classification accuracy, are standard and appropriate for assessing OOD detection performance.

**Other Comments Or Suggestions:**

See Weakness.

**Other Strengths And Weaknesses:**

Strengths:
1、 The paper introduces a novel approach combining adaptive prompts and contrastive learning, effectively addressing the challenges of few-shot OOD detection with limited labeled data.
2、 Extensive experiments and ablation studies demonstrate the method's robustness and effectiveness across multiple datasets and few-shot settings.

Weaknesses:

1、 Figure 1 is not referenced or discussed in the main text. The caption of Figure 7 contains a error: "right" and "left" are mistakenly reversed.
2、 The paper heavily relies on CLIP for feature extraction and prompt engineering. While this is a common choice, it raises the question of whether using alternative models for feature extraction could yield different results.
3、 The paper does not discuss the computational complexity or efficiency of the proposed method in comparison to existing methods.

**Questions For Authors:**

See Weakness.

**Relation To Broader Scientific Literature:**

The key contributions of this paper are well-situated within the broader scientific literature on out-of-distribution detection and few-shot learning, building upon and advancing several important concepts in these fields. The proposed AMCN leverages adaptive prompts and contrastive learning, which are both active areas of research in machine learning. The idea of using prompts to guide model learning is inspired by recent advancements in natural language processing, particularly in the context of prompting large language models for downstream tasks. This paper extends the concept of prompt learning to the domain of OOD detection, specifically targeting the challenging few-shot scenario where only limited labeled in-distribution samples are available. This approach is novel and addresses a significant gap in the literature, as most existing OOD detection methods rely on large amounts of labeled data and do not account for the diversity and scarcity of samples in few-shot settings.

**Theoretical Claims:**

No issues were found in the proof. The paper includes a theoretical claim regarding the projection of all feature vectors onto the unit hyper-sphere for cross-modal matching, which is essential for the multi-modal contrastive loss mechanism. This claim is supported by a detailed proof in Appendix A. The proof demonstrates that the L2 normalization applied to both image and text feature vectors ensures that all features lie on the surface of the unit hyper-sphere, thereby enforcing the use of cosine similarity as a dot product in the contrastive loss. The steps of the proof are logically structured and correctly derived, confirming that the features are constrained to the unit hyper-sphere as claimed.

---

> ### Author Rebuttal · Authors · 2025-03-30
>
> We sincerely thank Reviewer oFtr for the valuable comments and provide the following detailed responses to all weaknesses.
>
> ---
>
> ### **W1: Figure 1 is not referenced or discussed in the main text. The caption of Figure 7 contains a error: "right" and "left" are mistakenly reversed.**
>
> Thanks a lot for your valuable suggestion. Figure 1 should be referenced in Introduction. Figure 1(a) should be placed in the first paragraph of Introduction. Figure 1(b) should be placed in the second paragraph of Introduction (Line 81). Figure 1(c) should be placed in the third paragraph of Introduction (Line 96). We will revise the caption of Figure 7 in our revised version.
>
> ### **W2: The paper heavily relies on CLIP for feature extraction and prompt engineering. While this is a common choice, it raises the question of whether using alternative models for feature extraction could yield different results.**
>
> Thanks a lot for your valuable suggestion. 1) Since all the compared methods utilize CLIP to extract features, we follow them to use CLIP for fair comparison. 2) For many other feature encoders, they include more complex calculations, which might make the designed OOD detector time-consuming. 3) The CLIP network can effectively extract the visual and textual features under the few-shot setting. Therefore, we utilize CLIP in our paper.
>
>  ### **W3: The paper does not discuss the computational complexity or efficiency of the proposed method in comparison to existing methods.**
>
> We appreciate the reviewer’s comments regarding the computational complexity and efficiency of the proposed method. Compared to other methods, the training time and memory consumption are similar:
>
> |Method|Time for one iteration (s)|GPU Memory (MiB)|
> |-|-|-|
> |CoOp|0.54|20865|
> |LoCoOp|0.82|24180|
> |SCT|0.87|25267|
> |Ours|0.80|23946|
>
> We evaluate the time and memory consumption of our proposed method compared with other baselines in above table and the results show that our proposed method is relatively compute-efficient. The evaluation is conducted on a single GTX-4090
> GPU with a batch size as 64.

---

### Official Review · Reviewer_8TNR · 2025-03-09

**Overall Recommendation:** 5

**Summary:**

Out-of-distribution (OOD) detection is an important machine learning task. Few-shot OOD detection is an important yet challenging setting in OOD detection task, where only a few labeled ID samples are available. This paper proposes a novel and clear few-shot OOD detection model that considers an interesting and practical multi-diversity setting. Authors first generate adaptive prompts for ID classification. Then, authors generate an adaptive class boundary for each class by introducing a class-wise threshold to conduct adaptive ID alignment. Finally, authors design a prompt-guided ID-OOD separation module to control the margin between ID and OOD prompts for OOD detection.

**Claims And Evidence:**

Yes. The claims made in the submission supported by clear and convincing evidence.The experimental results can well show the effectiveness of the method.

**Essential References Not Discussed:**

No

**Experimental Designs Or Analyses:**

Yes. I checked the proof in the appendix, i.e., Section A. The proof is correct and reasonable. Authors show the superiority of their method in Table 1 and Figure 4.

**Methods And Evaluation Criteria:**

Yes. The proposed multi-diversity few-shot OOD detection method is practical and novel.

**Other Comments Or Suggestions:**

It would be better if authors could provide some failure examples of few-shot OOD detection.

**Other Strengths And Weaknesses:**

Strengths:

1. The idea  is interesting. This paper considers various distributions between different classes in the few-shot OOD detection task. It might provide some inspiration for the task of few-shot learning. Besides, the proposed method is novel and sound.

2. Based on one ID dataset (mageNet-1k) and four OOD datasets (Texture, Places, SUN and iNaturalist), authors conduct many experiments to verify the effectiveness of the proposed method. The performance of the proposed method AMCN is state-of-the-art. The corresponding ablation studies show the effectiveness of three modules (adaptive prompt generation, prompt-based multi-diversity distribution learning and prompt-guided OOD detection).

3. The overall structure of the proposed method is clear and the paper is easy to follow. In the experimental section, this performance analysis is reasonable.

4. Authors provide available codes, and the proposed method seems reproducible. Besides, the proposed method and uploaded code well match.

Weaknesses:

1. A little typo in Figure 1 caption. “(c) Brief framework of our method.” should be “(d) Brief framework of our method.”

2. T-SNE visualization in Figure 3 is based on which dataset? The sentence “As shown in Figure 3, different classes have distinct diversity” should be removed to the first paragraph of Section 3.2.

3.In the paragraph above Equation (7), “Based on (7)”, is this Equation (7) or Equation (6)? Is it a typo? I think it should be Equation (6).

**Questions For Authors:**

Please address the weakness. I am willing to raise the score

**Relation To Broader Scientific Literature:**

This paper considers various distributions between different classes in the few-shot OOD detection task. It might provide some inspiration for the task of few-shot learning.

**Theoretical Claims:**

N/A

---

> ### Author Rebuttal · Authors · 2025-03-30
>
> We sincerely thank Reviewer 8TNR for the valuable comments and provide the following detailed responses to all weaknesses.
>
> ---
>
> ### **W1: A little typo in Figure 1 caption. “(c) Brief framework of our method.” should be “(d) Brief framework of our method.”**
>
> Thanks a lot for your valuable suggestion. We will revise Figure 1 caption in our revised version.
>
> ### **W2: T-SNE visualization in Figure 3 is based on which dataset?**
>
> We appreciate the reviewer’s comments regarding T-SNE visualization in Figure 3. The dataset of T-SNE visualization in Figure 3 is based on the ImageNet-1k dataset.
>
> ### **W3: The sentence “As shown in Figure 3, different classes have distinct diversity” should be removed to the first paragraph of Section 3.2.**
>
> Thank you for your constructive suggestion. We will revise it in our revised version.
>
> ### **W4: In the paragraph above Equation (7), “Based on (7)”, is this Equation (7) or Equation (6)? Is it a typo?**
>
> Thanks a lot for your valuable comment. Yes. It should be Equation (6). We will revise it in our revised version.
>
> ### **W5: It would be better if authors could provide some failure examples of few-shot OOD detection.**
>
> In Few shot OOD, its paramount to generate ID and OOD prompts.
> Therefore, the dataset providing the labels (ID dataset) is key. It should be diverse enough.
> This is the case of ImageNet1K which is used in the benchmarks.
>
> However, if we take a more specific dataset as ID, such as PLACES, which is less diverse, then the prompt engineering is less effective:
>
> |Shot|FPR95($\downarrow$)|AUROC($\uparrow$)|
> |-|-|-|
> |1|44.21|88.50|
> |8|43.75|89.74|
>
> Here, the Places dataset (less classes) is treated as ID set and the ImageNet-1k dataset as OOD set (reversed compared to the benchmarks). We can find that when we switch the Places dataset and the ImageNet-1k dataset, the performance improvement is reduced.

---

> > ### Comment · Reviewer_8TNR · 2025-04-03
> >
> > Thank you for the responses. The authors addressed all of my concerns. After considering the comments from other reviewers and authors' rebuttals, I believe that this paper is an excellent work. So I raised my ratings.

---

> > > ### Author Response · Authors · 2025-04-05
> > >
> > > We greatly appreciate your enthusiastic and uplifting support for our efforts! It’s a true privilege to learn that you see our work as significant and influential. Realizing that our contributions strike a chord with others in the community is deeply inspiring. Once more, we extend our sincere gratitude for your insightful feedback and steadfast encouragement.

---

### Official Review · Reviewer_1bUn · 2025-03-09

**Overall Recommendation:** 4

**Summary:**

This work introduces a new method for few-shot out-of-distribution (OOD) detection. Unlike previous approaches that largely overlook the diverse characteristics among different classes, the proposed method constructs both in-distribution (ID) and OOD prompts and designs multiple contrastive losses to learn a better separation boundary. Experiments on multiple datasets demonstrate significantly improved performance compared to existing methods.

**Claims And Evidence:**

The claimed contributions are validated by ablation experiments.

**Essential References Not Discussed:**

No

**Experimental Designs Or Analyses:**

The experimental design is reasonable, and enough ablation analyses are provided.

**Methods And Evaluation Criteria:**

The proposed method looks sound, and the evaluation metrics are proper.

**Other Comments Or Suggestions:**

the equation following Equation 3 is not numbered. Based on the context in the later sections, it seems to represent 𝐿1, not 𝐿𝑐.

**Other Strengths And Weaknesses:**

Strengths:
The work is well-motivated, and a reasonable solution is presented to address the class diversity issue.

The experimental results are strong and clearly show the effectiveness of the proposed method.

Weaknesses:
Prompt learning is widely explored in the community. The idea of using learnable prompts for both ID and OOD classes is straightforward. However, the design of contrastive losses between different sets of prompts adds a meaningful contribution.

The proposed method involves a large number of hyperparameters. The authors should conduct a sensitivity analysis to understand the impact of these parameters on model performance.

In the methodology section, there is a lack of discussion regarding the rationale behind certain technical design choices. For example, it is unclear how class-wise thresholding addresses the issue of sample diversity—this should be further elaborated.

**Questions For Authors:**

The authors should conduct a sensitivity analysis to understand the impact of these parameters on model performance.

**Relation To Broader Scientific Literature:**

Out-of-distribution detection is of great interest to the wide computer vision community, it has much value in real-world scenarios, e.g., autonomous driving.

**Theoretical Claims:**

There is no theoretical proof in this work.

---

> ### Author Rebuttal · Authors · 2025-03-30
>
> We sincerely thank Reviewer 1bUn for the valuable comments and provide the following detailed responses to all weaknesses.
>
> ---
>
> ### **W1: Prompt learning is widely explored in the community. The idea of using learnable prompts for both ID and OOD classes is straightforward. However, the design of contrastive losses between different sets of prompts adds a meaningful contribution.**
>
> We appreciate the reviewer’s comments regarding learnable prompts and contrastive losses. Firstly, we combine $P$ learnable ID prefixes and the label name to generate the learnable ID prompts (LIPs). Also, we generate $S$ labelfixed OOD prompts (LFOPs) by introducing OOD labels from other datasets that are disjoint with the ID label set. Since the introduced OOD labels are often limited, we explore $Z$ label-adaptive OOD prompts (LAOPs) for each ID prompt. Besides, we align the image features and ID prompt features by a prompt-guided contrastive loss for ID classification.
>
> ### **W2: The authors should conduct a sensitivity analysis to understand the impact of these parameters on model performance.**
>
> Thanks a lot for your valuable suggestion. Due to the page limitation, we have analyzed some significant parameter in Figure 5.
> We are using hyperparameters $\tau_0, \ldots, \tau_5$ in the tool. We are reporting here the sensitivity analysis towards these hyperparameters on of the benchmark SUN, in the eight-shot setting:
>
> |$\tau_0$|FPR95($\downarrow$)|AUROC($\uparrow$)|
> |-|-|-|
> |0.1|23.46|95.71|
> |**0.2**|**23.17**|**95.89**|
> |0.3|23.32|95.82|
>
> |$\tau_1$|FPR95($\downarrow$)|AUROC($\uparrow$)|
> |-|-|-|
> |0.40|23.32|95.83|
> |**0.45**|**23.17**|**95.89**|
> |0.50|23.53|95.74|
>
> |$\tau_2$|FPR95($\downarrow$)|AUROC($\uparrow$)|
> |-|-|-|
> |0.4|23.29|95.72|
> |**0.5**|**23.17**|**95.89**|
> |0.6|23.35|95.83|
>
> |$\tau_3$|FPR95($\downarrow$)|AUROC($\uparrow$)|
> |-|-|-|
> |0.50|23.40|95.58|
> |**0.55**|**23.17**|**95.89**|
> |0.60|23.29|95.74|
>
> |$\lambda$|FPR95($\downarrow$)|AUROC($\uparrow$)|
> |-|-|-|
> |0.2|23.98|94.35|
> |**0.3**|**23.17**|**95.89**|
> |0.4|23.67|95.12|
>
> ### **W3: In the methodology section, there is a lack of discussion regarding the rationale behind certain technical design choices. For example, it is unclear how class-wise thresholding addresses the issue of sample diversity—this should be further elaborated.**
>
> We also appreciate the reviewer’s comments regarding the class-wise threshold. In the fixed setting, we do not update the threshold during training. Since there are different diversities in different classes, we want to learn different thresholds for different classes. Different diversities correspond to different distributions. In Section 3.3, we learn the distribution of each class in "Learning distribution" to obtain the corresponding diversity information. In "Intra-class distribution normalization", to fully learn the intra-class distribution of ID samples for better classification, we independently normalize the distribution for each class by $L_I^1$. In "Inter-class distribution normalization", we balance the distributions of all the classes by $L_I^2$. By learning the class-wise threshold and data distributions (intra-class distribution and inter-class distribution), we can obtain an adaptive classification decision boundary for each class, which effectively reduces the negative impact of different diversities on the challenging few-shot OOD detection task.
>
> ### **W4: the equation following Equation 3 is not numbered. Based on the context in the later sections, it seems to represent $L_1$, not $L_C$.**
>
> We thank the reviewer for the insightful suggestions.We will add the equation number in the final version. In fact, it is $L_C$, the $L_1$ is defined in Eq. (9).

---

> > ### Comment · Reviewer_1bUn · 2025-04-07
> >
> > I thank the authors for their comprehensive and detailed responses to my questions. After reading their rebuttal, I believe the authors have addressed all the questions I raised. I believe the manuscript meets the acceptance standard, and I will maintain my score.

---

### Official Review · Reviewer_J7Ff · 2025-03-10

**Overall Recommendation:** 4

**Summary:**

This paper proposes to address the novel and challenging task, multi-diversity few-shot OOD detection. Unlike the previous methods that ignore the distinct diversity between different classes in the few-shot OOD detection task, this paper presents a novel network AMCN. The proposed method first transposes ID prompts into OOD prompts by semantically concatenating ID prompts with OOD suffixes. The semantic concatenation can generate many negative prompts to guide prompt learning in the few-shot OOD detection task with multi-diversity setting. Finally, the paper develops an ID-OOD separation module to control the margin between ID  and OOD prompt features by a carefully-designed loss.

**Claims And Evidence:**

The viewpoints are well verified. Specifically, the paper presents three modules to handle multi-diversity few-shot OOD detection. In the ablation study, this paper provides many ablation results to support the claims in Methodology.

**Essential References Not Discussed:**

No.

**Experimental Designs Or Analyses:**

Yes, I have checked the soundness of any experimental designs and analyses.

**Methods And Evaluation Criteria:**

Yes. Both the proposed method and evaluation criteria make sense for the problem of few-shot OOD detection at hand.

**Other Comments Or Suggestions:**

In Figure 4 and Figure 7, the sub-figures are too crowded.

**Other Strengths And Weaknesses:**

Strengths:
1. This paper proposes an expressive and novel Adaptive Multi-prompt Contrastive Network, especially in the few-shot OOD detection task. This paper utilizes a certain number of images with different diversity from each class for training, and conduct OOD detection on the whole testing dataset. Different from previous works that only learn ID prompts for training, this paper constructs ID and OOD prompts for each class to fully understand the images. The setting is very interesting, and the proposed method makes sense.
2. The proposed method is effective. This paper first generates adaptive prompts (learnable ID prompts, label-fixed OOD prompts and label-adaptive OOD prompts). Also, this paper learns an adaptive class boundary for each class by introducing a class-wise threshold. Finally, this paper proposes a prompt-guided ID-OOD separation module to control the margin between ID and OOD prompts.
3. Extensive comparisons with state-of-the-arts are conduct in the experiment section. Corresponding results demonstrate that the proposed approach can achieve significant performance on four benchmarks under the few-shot OOD detection setting.
4. The motivation of exploring different distributions between different classes in the few-shot OOD detection task and the proposed method are impressive. It is very enlightening to the research community and will inspire more future research. The paper is well-prepared and easy to understand.
Weaknesses:
1. Authors should move the definition of c (“$c \in \{1,...,C\}$ is the corresponding class of $f_x^i$”) to the third paragraph of Section 3.2.
2. In Eq. (7), $O \cdot \mathcal{M}_c^{pse}(t)$ should be $O \cdot \mathcal{M}_c^{pse}(t-1)$?
3. In Table 4, what is the fixed threshold?

**Questions For Authors:**

I don't have any other specific questions. Please answer the questions in the weaknesses.

**Relation To Broader Scientific Literature:**

Authors make some progress and promote the field of few-shot OOD detection. Especially, authors solve the significant challenges about the distinct diversity between different classes. The experiment section demonstrates impressive performance of the proposed method.

**Theoretical Claims:**

Yes, I have checked the correctness for theoretical claims. This paper clearly presents the motivation of the proposed multi-diversity few-shot OOD detection setting. Three modules in Figure 2 and the Methodology section make sense for solving three challenges: ID classification, adaptive ID alignment, and OOD detection.

---

> ### Author Rebuttal · Authors · 2025-03-30
>
> We sincerely thank Reviewer J7Ff for the valuable comments and provide the following detailed responses to all weaknesses.
>
> ---
>
> ### **W1: Authors should move the definition of c to the third paragraph of Section 3.2.**
>
> Thanks a lot for your valuable suggestion. We will revise it in our revised version.
>
> ### **W2: In Eq. (7), $O \cdot \mathcal{M}_c^{pse}(t)$ should be $O \cdot \mathcal{M}_c^{pse}(t-1)$?**
>
> We thank the reviewer for the insightful suggestions. Yes. It should be $O \cdot \mathcal{M}_c^{pse}(t-1)$. We will revise it in our revised version.
>
> ### **W3: In Table 4, what is the fixed threshold?**
>
> We appreciate the reviewer’s comments regarding the fixed threshold in Table 4. It is the initial threshold. In the fixed setting, we do not update the threshold during training. Since there are different diversities in different classes, we want to learn different thresholds for different classes. Different diversities correspond to different distributions. In Section 3.3, we learn the distribution of each class in "Learning distribution" to obtain the corresponding diversity information. In "Intra-class distribution normalization", to fully learn the intra-class distribution of ID samples for better classification, we independently normalize the distribution for each class by $L_I^1$. In "Inter-class distribution normalization", we balance the distributions of all the classes by $L_I^2$.
>
> ### **W4: In Figure 4 and Figure 7, the sub-figures are too crowded.**
>
> We thank the reviewer for the insightful suggestions. We will redesign them in our final version.

---

### Decision · Program_Chairs · 2025-05-01

**Decision:**

Accept (spotlight poster)

**Comment:**

This paper proposes Adaptive Multi-prompt Contrastive Network (AMCN) for few-shot out-of-distribution detection. It tackles class diversity and data scarcity by engineering ID/OOD prompts using CLIP and introducing contrastive learning losses to adaptively separate ID and OOD features.

All reviewer believe that the paper addresses a meaningful and challenging problem with a novel and well-motivated design. All reviewers affirm its sound methodology, strong experimental results, and contributions to few-shot OOD detection. The code is available, and the method is reproducible and well-executed.

Concerns are mostly minor from reviewers: partially lack of sensitivity analysis, insufficient discussion of design rationale, and dependency on CLIP. One reviewer questions generalization across backbones and notes some clarity and formatting issues, but none are critical flaws.